# A Novel Double-Diamond Microreactor Design for Enhanced Mixing and Nanomaterial Synthesis

**DOI:** 10.3390/mi16091058

**Published:** 2025-09-18

**Authors:** Qian Peng, Guangzu Wang, Chao Sheng, Haonan Wang, Yao Fu, Shenghong Huang

**Affiliations:** 1CAS Key Laboratory of Mechanical Behavior and Design of Materials, School of Engineering Science, University of Science and Technology of China, Hefei 230026, China; pengqian@mail.ustc.edu.cn (Q.P.); whn@mail.ustc.edu.cn (H.W.); 2Institute of Advanced Technology, University of Science and Technology of China, Hefei 230031, China; wgz@mail.ustc.edu.cn (G.W.); sc0221@mail.ustc.edu.cn (C.S.); 3Hefei National Research Center for Physical Sciences at the Microscale, iChEM (Collaborative Innovation Center of Chemistry for Energy Materials), CAS Key Laboratory of Urban Pollutant Conversion, Anhui Province Key Laboratory of Biomass Clean Energy, University of Science and Technology of China, Hefei 230026, China

**Keywords:** microreactor, chaotic mixing, nanoparticle synthesis, computational fluid dynamics, passive micromixer, process intensification

## Abstract

This study introduces the Double-Diamond Reactor (DDR), a novel planar passive microreactor designed to overcome the following conventional limitations: inefficient mass transfer, high flow resistance, and clogging. The DDR integrates splitting–turning–impinging (STI) hydrodynamic principles via CFD-guided optimization, generating chaotic advection to enhance mixing. Experimental evaluations using Villermaux–Dushman tests showed a segregation index ($X_{s}$) as low as 0.027 at 100 mL·min^−1^, indicating near-perfect mixing. In BaSO_4_ nanoparticle synthesis, the DDR achieved a 46% smaller average particle size (95 nm) and narrower distribution ($\sigma_{g}=1.27$) compared to reference designs (AFR-1), while maintaining low pressure drops (<20 kPa at 60 mL·min^−1^). The DDR’s superior performance stems from its hierarchical flow division and concave-induced vortices, which eliminate stagnant zones. This work demonstrates the DDR’s potential for high-throughput nanomaterial synthesis with precise control over particle characteristics, offering a scalable and energy-efficient solution for advanced chemical processes.

## 1. Introduction

Microreactors have revolutionized chemical processes by enabling rapid mixing and precise control over reaction conditions, particularly in nanomaterial synthesis. However, their performance is often limited by inefficient mass transfer and high energy consumption under laminar flow regimes [1]. At microscale dimensions, where laminar flow regimes dominate and mass transport is principally governed by molecular diffusion, these systems face unique mixing challenges. To address this, microreactors employ either passive or active mixing strategies [2]. Active systems utilize external energy inputs, such as acoustic waves, electric fields, or magnetic forces [3,4,5], while passive designs rely solely on fluid pressure and innovative channel geometries, including rectangular, zigzag, split-and-recombine, and circular configurations [2,6,7]. The passive approach offers distinct advantages in operational cost control and energy efficiency, making it particularly attractive for industrial production processes.

The performance of passive micromixing is crucial for designing microreactors tailored to specific chemical reactions [8]. Significant advances in this field include the square-wave microreactor developed by Shi et al. [6], which incorporates elliptical grooves along the sidewalls to suppress stagnation zones and promote fluid reutilization through Dean vortex generation at low Reynolds numbers. Similarly, Hu et al. [2] demonstrated that a multilayered polyimide zigzag microchannel configuration can simultaneously optimize mixing efficiency while minimizing pressure drop. Sheu et al. [7] further advanced the field by engineering two-dimensional staggered curved channels with tapered structures, where the synergistic combination of split-and-recombine geometries, secondary flows, and flow impingement significantly enhanced mixing performance. Another innovative approach by Mubashshir et al. [9] employed soft lithography to fabricate micro T-mixers that generate vortical flows at the junction between inlet and mixing channels, thereby improving mass transfer efficiency. Shang et al. [10] manufactured a novel microdevice called a single-chamber micromixer (SCM), which was designed to achieve good mixing at a low pressure drop condition, and the special chamber induced a strong oscillating flow, which is generated based on the Coanda effect. Li et al. [11] designed and compared four micromixers based on different mixing mechanisms, named wavy, circular, spiral, and baffle. The experimental results demonstrated that the baffle micromixer performed the best in micromixing. Wang et al. [12] investigated the chaotic characteristics in sudden convergent–divergent micromixers based on the Poincaré section and the Lyapunov exponent. The maximum mixing index was achieved at global chaos by varying the arrangement of the mixing. Collectively, these studies establish that strategically designed microstructures can substantially enhance mass transfer performance in microreactor systems.

Passive microreactors offer broad applicability across chemical processes due to their energy-efficient operation and straightforward implementation. Nevertheless, these systems exhibit several inherent limitations, including exponentially increasing pressure resistance at elevated flow rates and inadequate interfacial contact between fluid phases. While simplistic microreactor designs often demonstrate suboptimal mixing performance and mass transfer characteristics that hinder industrial-scale deployment, sophisticated architectures can overcome these challenges to enable consistent and homogeneous product formation. A prime example is the Corning Advanced Flow Reactor (AFR), which has achieved widespread industrial adoption through its exceptional gas–liquid two-phase mixing capability, achieving complete precursor mixing within remarkably short residence times of one minute [13].

The AFR system’s remarkable performance stems from its innovative heart-shaped unit design, which generates converging–diverging flow patterns that intensify interfacial contact [14,15]. This configuration has demonstrated 93% conversion efficiency at flow rates of 30 mL/h [16], with reaction completion occurring within the first row of heart-shaped elements during the ozonolysis processes of Sudan Red 7B dye [17]. Optimization studies reveal that geometric parameters significantly influence system performance, with a corner height of 0.4 mm and a 90° angle providing optimal mixing efficiency while maintaining bubble fragmentation control across operational flow ranges [18,19]. The technology’s scalability is evidenced by its successful scale-up pathway from spiral microreactors through the Low Flow Reactor (LFR) to the full AFR system, achieving a 700-fold production increase without mass transfer compromise [20].

However, comprehensive CFD analyses have identified persistent challenges, including stagnant zones that reduce momentum transfer efficiency [21] and flow pattern sensitivity to minor structural variations. While the AFR system represents a significant advancement over its predecessor LFR—supporting flow rates up to 100 mL/min compared to LFR’s 10 mL/min limit [22]—these findings underscore the need for continued innovation in microreactor design. An optimal architecture must simultaneously achieve three critical objectives: (1) the elimination of stagnant regions, (2) a reduction of pressure resistance without mixing efficiency sacrifice, and (3) a maximization of interfacial contact area. Such advancements would enable precise control over nanoparticle characteristics (size, distribution, and magnetic properties) [23] while maintaining the rapid heat and mass transfer advantages characteristic of microreactor systems [24], ultimately facilitating high-quality product manufacturing.

This study proposes the periodic planar Double-Diamond Reactor (DDR), architecturally optimized via CFD. Departing from conventional designs, it integrates (1) bifurcated flow splitters for multi-stream division; (2) serpentine acute-angle channels enabling continuous sharp turning; and (3) localized concave impinging zones inducing fluid folding and collision. The rhomboidal cellular network orchestrates recursive STI sequences (splitting–turning–impinging), where acute turning amplifies stretching before localized impinging drives chaotic mixing.

To validate the CFD-optimized structure of the microreactor, we executed a tripartite verification strategy: First, leveraging the same CFD framework used for reactor design, we simulated velocity fields and concentration gradients to decode mixing mechanisms. Subsequently, quantitative benchmarks against an industry-standard AFR-like design via Villermaux–Dushman tests revealed significantly lower segregation indices. Critically, continuous BaSO_4_ synthesis at high-throughput/low-resistance conditions confirmed superior performance: smaller nanoparticles with narrower distribution, unequivocally validating the STI mechanism’s efficacy and advancement.

## 2. Numerical and Experimental Methods

### 2.1. Basic Configurations of New Microreactor Design

The three planar microchannel configurations—the new Double Diamond Reactor (DDR, Figure 1a) and two reference designs (AFR-1 and AFR-2, Figure 1b,c)—each feature multiple mixing element rows with dual reagent inlets and single effluent outlets, maintaining comparable unit holdup volumes while implementing distinct mixing strategies. The DDR introduces an innovative architecture with double-diamond-shaped overlapping outlines incorporating seven rhomboid inner splitters, each featuring concave foreparts to induce controlled fluid impacts. This sophisticated design implements a hierarchical splitting system: wider splitters near the inlet–outlet regions initially divide the flow into two streams, while narrower mid-channel splitters further separate these into four substreams. The subsequent geometric contraction directs substreams into turning–impinging zones for chaotic mixing, creating systematic lateral perturbations that significantly enhance mass transfer.

In contrast, the reference AFR-1 (Figure 1b) employs a simplified U-shaped obstacle paired with a single turbulent cylinder—the former redirects flow through sharp turns while the latter provides basic flow recombination—implemented in reduced channel dimensions to assess the inlet size effects. Its modified counterpart, AFR-2 (Figure 1c), maintains the DDR’s channel height and inlet width while preserving the U-cylinder configuration, serving as a direct performance benchmark that isolates geometric influences from scale factors. This carefully structured comparison enables a precise evaluation of how splitting complexity (DDR vs. U-cylinder designs) and dimensional scaling (AFR-1 vs. AFR-2) independently affect the mixing efficiency and pressure drop characteristics.

### 2.2. Computational Fluid Dynamics (CFD) Simulation

To obtain details on the flow field and component concentration, CFD simulation can be used for analyzing local mixing and illustrating the mechanism by solving the continuity, momentum, energy, and mass equations of conservation.(1)Continuityequation:∇·u=0(2)$$\mathrm{Momentum}\mathrm{equation}:\rho \left[\frac{\partial u}{\partial t}+(u\cdot \nabla )u\right]=-\nabla p+\mu \nabla^{2}u$$(3)$$\mathrm{Energy}\mathrm{equation}:\rho c_{p}\left[\frac{\partial T}{\partial t}+(u\cdot \nabla )T\right]=\nabla \cdot (k\nabla T)+q$$(4)$$\mathrm{Species}\mathrm{equation}:\frac{\partial (\rho_{i})}{\partial t}+\nabla \cdot \left(\rho_{i}u\right)=\nabla \cdot \left(D_{i}\nabla \rho_{i}\right)+r_{i}$$
where u is the velocity vector, ρ is the fluid density, *t* is the time, μ is the fluid viscosity, *p* is the pressure, $c_{p}$ is the specific heat of the species, *T* is the temperature, k is the thermal conductivity, q is the internal heat source term, $\rho_{i}$ is the mass concentration of species *i*, $D_{i}$ is the diffusion coefficient of species *i*, and $r_{i}$ is the formation rate of species *i*. Unsteady and incompressible Newtonian liquid conditions are assumed to solve the above equations.

For the simulation of fluid mixing, water and nitric acid entered the microchannel through the left and right inlets, while the mass fraction of water in the left inlet passage was marked as zero, and that in the right inlet passage was marked as one. The density of water is set to 998.2 $kg\cdot m^{-3}$, and the dynamic viscosity of water is set to 1.003 $\times \;10^{-3}\;kg\cdot m^{-1}\cdot s^{-1}$. The density and dynamic viscosity of nitric acid are 1504 $kg\cdot m^{-3}$ and 1.1 $\times \;10^{-3}\;kg\cdot m^{-1}\cdot s^{-1}$, respectively. 

The finite-volume method is utilized to discretize all three equations of conservation, and the simulation is performed using the software ANSYS Fluent 19.2 operated in a double-precision model. The 3D geometry of the reactors was constructed using CAD 2021. Then, the reactors were meshed into structured meshes with hexahedral elements generated by the ICEM software 19.2, and the mesh details are shown in Figure 2. In the CFD simulations, non-conformal meshes typically require greater computational resources. Given the same number of mesh elements, a more conformal mesh enables faster convergence within fewer iterations. This yields flow field and mixing results that are more physically accurate and representative of real-world behavior. Therefore, we have implemented appropriate grid refinement in the near-wall, collision-mixing, and shear-mixing regions, as shown in Figure 2a,b.

To acquire the mesh-independent solution, five different numbers of elements, ranging from 1,358,997 to 9,512,979, are built for the independent test, as shown. It is clear that with the increase in mesh elements, the pressure drop of the simulated unit, the velocity distribution curve, and the normalized concentration (range 0–1) distribution curve at the outlet tangent line are gradually converging to a fixed value or curve, as shown in Figure 2c–e. And a grid of 6794985 is selected in consideration of both computational costs and accuracy. The iterative procedure is repeated until all equations meet the convergence criterion of $10^{-6}$ in all cases, to ensure solution accuracy.

### 2.3. Parallel Competing Reaction System of Villermaux–Dushman Test

The iodide–iodate test reaction [25,26], which is also called the Villermaux–Dushman reaction, has been most commonly employed to acquire quantitative data to evaluate the mixing efficiency for a microreactor due to its ease of measurement and adjustable sensitivity [27]. The reaction system couples the neutralization and comproportionation of iodide and iodate ions to form iodine [25]. The side product of triiodide is easy to quantify by using UV-visible spectroscopy. This approach is based on the following three chemical reactions:(5)$$H_{2}BO_{3}^{-}+H^{+}\to H_{3}BO_{3}\left(\mathrm{quasi}-\mathrm{instantaneous}\right)$$(6)$$5I^{-}+IO_{3}^{-}+6H^{+}\to 3I_{2}+3H_{2}O\left(\mathrm{very}\;\mathrm{fast}\right)$$(7)$$I_{2}+I^{-}⇄I_{3}^{-}\left(\mathrm{instantaneous}\;\mathrm{equilibrium}\;\mathrm{reaction}\right)$$

In the system, Reactions (5) and (6) compete for hydrogen ions, neutralization is quasi-instantaneous, and Reaction (6) is much slower according to the kinetic parameters (Table 1). If a multiphase fluid mixes perfectly, the acid solution will be completely dispersed and consumed by the borates. Under ideal mixing conditions, all protons in the acid will solely react with the base due to extremely fast kinetics, and thus, no iodine will be generated in the microchannel. However, if the mixing proceeds poorly, local excess protons will react with iodide and iodate ions to form iodine, which will further react with iodide ions to yield triiodide ions, according to Reactions (6) and (7). Generally, the segregation index $X_{s}$ is adopted to quantify the micromixing performance, which is defined as follows [28]:


(8)
$$X_{s}=\frac{Y}{Y_{ST}}$$


In Equation (8), Y represents the ratio of the $H^{+}$ mole number consumed by reaction 6 to the total $H^{+}$ mole number in the mixing solution. $Y_{ST}$ denotes the value of Y in the total segregation when the micromixing process is infinitely slow.(9)$$Y=\frac{2(V_{A}+V_{B})(\left[I_{2}\right]+[I_{3}^{-}])}{V_{B}{\left[H^{+}\right]}_{0}}$$(10)$$Y_{ST}=\frac{6{\left[IO_{3}^{-}\right]}_{0}}{6{\left[IO_{3}^{-}\right]}_{0}+{\left[H_{2}BO_{3}^{-}\right]}_{0}}$$
where $V_{A}$ and $V_{B}$ are the volumetric flow rates of mixtures *A* and *B*. In this study, mixture *A* contains $H_{2}BO_{3}^{-}$, $I^{-}$, and $IO_{3}^{-}$, and mixture B is a sulfuric acid solution. $\left[I_{2}\right]$ and $\left[I_{3}^{-}\right]$ represent the concentrations of $I_{2}$ and $I_{3}^{-}$ ions in the reaction effluent, respectively, while ${\left[H^{+}\right]}_{0}$ is the initial concentration of $H^{+}$ ions in the sulfuric acid solution. In addition, ${\left[IO_{3}^{-}\right]}_{0}$ and ${\left[H_{2}BO_{3}^{-}\right]}_{0}$ denote the initial concentrations of $IO_{3}^{-}$ and $H_{2}BO_{3}^{-}$, respectively, in the buffer solution. The segregation index $X_{s}$ ranges from zero to one, and a value of zero indicates perfect quick micromixing, while a value of one indicates complete segregation. To obtain the data detail of the concentration of $I_{3}^{-}$ ions, measurement by a UV-Vis spectrophotometer at 353 nm is adopted in the experiment, while the concentration of $I_{2}$ can be calculated through an equilibrium constant $K_{B},$ as follows [31,32]:(11)$$K_{B}=\frac{\left[I_{3}^{-}\right]}{\left[I_{2}\right]\left[I^{-}\right]}$$(12)$${{log}_{10}K}_{B}=\frac{555}{T}+7.355-2.575{log}_{10}T$$

### 2.4. Synthesis of BaSO_4_ Nanoparticles

A schematic diagram of the experimental setup for the continuous generation of BaSO_4_ nanoparticles is shown in Figure 3. Aqueous BaCl_2_ and Na_2_SO_4_ solutions are pumped into the microreactor by two syringe pumps (LST02-1B, Hyuweiy (Fluid Equipment Co., Ltd. Beijing, China). The total quantity of flow ranges from 20 to 100 $mL\cdot {min}^{-1}$, and the flux ratio of the two reactants remains at 1. After the process of reaction in the microreactor approaches steady conditions, the fluid containing the nanoparticles is collected in a beaker. In the laboratory, the product is centrifuged and dried before being collected in an airtight container. To avoid different precursor concentrations affecting each other and preventing clogging, the reactor was flushed with deionized water twice after the experiment.

### 2.5. Materials

Potassium iodide (KI, >99 wt%, Aladdin (Shanghai, China)), potassium iodate ($KIO_{3}$, >99 wt%, Aladdin), iodine ($I_{2}$, >99 wt%, Aladdin), orthoboric acid ($H_{3}{BO}_{3}$, >99%, Aladdin), sodium hydroxide (NaOH, >97 wt%, Siopharm Chemical Reagent Co., Ltd. (Shanghai, China)), barium chloride (BaCl_2_, >99 wt%, Macklin Biochemical Technology Co., (Shanghai, China)), and sulfuric acid ($H_{2}{SO}_{4}$, >98 wt%, Aladdin) were purchased and used without further purification. The deionized (DI) water (resistivity of 18.2 MΩ·cm) used in the experiments was purified by a Milli-Q Direct8/16 (Merck Ltd. (Chengdu, China)) ultrapure water system.

## 3. Results and Discussion

### 3.1. Flow Characteristics and Mixing Behavior Based on CFD Analysis

While microchannel flow organization fundamentally determines mixing performance, conventional experimental methods often fail to capture the intricate flow details governing these processes. To address this limitation, we employed computational fluid dynamics (CFD) to conduct a systematic comparative analysis of the flow characteristics and pattern evolution in three distinct microchannel configurations. This approach enables precise identification of structural features that optimally enhance mixing efficiency while minimizing flow resistance, providing critical insights for the flow characteristics of the new proposed microreactor configuration.

#### 3.1.1. Flow Field

The distinct flow patterns and mixing mechanisms in each microreactor configuration were elucidated through detailed velocity contour, streamline analysis, and vorticity contour (Figure 4). In the AFR-1 design (Figure 4b,e,h), the characteristic U-shaped obstacle creates pronounced flow separation, generating two counter-rotating vortices with peak velocities of 0.32 $m\cdot s^{-1}$ at the inlet–outlet regions. This configuration produces significant velocity gradients (0.32 to 0.1 $m\cdot s^{-1}$) and distinct stationary zones, with the expanding side channels inducing additional expansion vortices that may compromise mixing uniformity. In contrast, the DDR configuration (Figure 4a,d,g) demonstrates superior flow organization through its innovative structural features:

(1)Concave splitter design: The strategically positioned concave elements at each splitter forepart create controlled fluid impacts, generating intense lateral perturbations that enhance interfacial contact without creating large stagnant regions;(2)Hierarchical flow division: The multi-stage splitting system (initial division into two streams by wider splitters, followed by four-way splitting via narrower elements) maintains coherent flow structures while progressively increasing the interfacial area;(3)Dynamic velocity modulation: The design achieves smooth velocity transitions (from 0.32 to 0.1 $m\cdot s^{-1}$) through gradual geometric contractions, avoiding the abrupt changes observed in AFR-1;(4)Continuous impinging convergence: The mirror-symmetric bottom section directs substreams into folding–impinging zones for chaotic mixing, with confluent structures maintaining momentum while preventing flow separation.

The AFR-2 variant (Figure 4c,f,i), while exhibiting reduced maximum velocities (0.22 $m\cdot s^{-1}$) due to its expanded inlet dimensions compared with AFR-1, retains the fundamental vortex generation mechanism of AFR-1. However, the weaker vortical structures in AFR-2 compared to those in AFR-1 confirm that the vortex formation in AFR units is predominantly dictated by the inlet flow velocity, making the mixing performance highly sensitive to the feed rate control. Key advantages of the DDR architecture become particularly evident when examining:(1)The absence of large stagnant zones that plague both AFR designs;(2)More uniform velocity distributions throughout the mixing cell;(3)Sustained lateral perturbations that persist through multiple splitting–turning–impinging convergence cycles;(4)The ability to maintain mixing efficiency while reducing pressure losses.

This comparative analysis demonstrates how DDR’s structural innovations—particularly its concave splitters and optimized splitting hierarchy—fundamentally transform flow patterns to achieve superior mixing performance compared to conventional U-obstacle designs.

#### 3.1.2. Concentration Field

To elucidate the coupling between flow patterns and concentration distributions within the microchannels, we systematically analyzed the mass fraction contour profiles. Figure 5a–c present the longitudinal section mass fraction distributions for DDR, AFR-1, and AFR-2 configurations at a fixed total flow rate of 20 $mL\cdot {min}^{-1}$, while Figure 5d–f comparatively display their concentration profiles across varying flow rates. Furthermore, we introduce the mixing index (*MI*) as a quantitative method for evaluating the mixing efficiency of the microreactor. This index is calculated based on the standard deviation of species concentration across the microreactor cross-sections. In comparison to the qualitative evaluation of mixing performance using concentration contour maps, this approach provides clear advantages. The mathematical formulation of *MI* is given by:(13)$$MI=1-\frac{\sigma }{\sigma_{max}}$$(14)$$\sigma =\sqrt{\frac{{\sum_{i=1}^{N}\left(c_{i}-\bar{c}\right)}^{2}}{N}}$$
where *σ* represents the standard deviation of the local concentration distribution across the cross-sectional plane, and $\sigma_{max}$ corresponds to the maximum possible concentration variance, defined as 0.5 for this binary mixing system. The parameter *N* denotes the total number of discrete sampling points within the analyzed plane, while $c_{i}$ indicates the instantaneous concentration value at the *i*-th sampling location. The term $\bar{c}$ signifies the ideal uniform mixing concentration, which equals 0.5 for perfectly mixed conditions. The mixing index (*MI*) is dimensionless and bounded between 0 and 1 (or 0% to 100%), where *MI* = 0 signifies completely segregated flows and *MI* = 1 represents ideal homogeneous mixing.

The mass fraction distribution is visualized using a color-coded scheme where the right inlet stream (red) is assigned a value of 1 and the left inlet stream (blue) 0, with perfect mixing represented by a homogeneous green color at 0.5 (Figure 5a–f). The temporal evolution of mixing is clearly demonstrated in Figure 5d–f, showing the progression from initially segregated streams (0 and 1) to nearly homogeneous conditions (approaching 0.5) downstream. Quantitative analysis reveals two key trends: (1) mixing efficiency improves progressively through successive reactor cells, and (2) higher volumetric flow rates enhance mixing quality, as evidenced by the more rapid transition to green hues in the contour plots.

The DDR’s superior performance stems from its innovative concave structures that create controlled flow impacts, generating vortices that promote chaotic advection (Figure 4a,d,g). This design enables three synergistic mixing mechanisms: (i) periodic stretching and folding of fluid layers, (ii) enhanced interfacial area through multi-stage splitting (Figure 1a), and (iii) transverse flow perturbations that disrupt laminar flow patterns. The comparative visualization in Figure 5 demonstrates that DDR achieves more complete mixing within fewer cells than AFR-1 or AFR-2, particularly at elevated flow rates where its chaotic advection mechanisms become most effective.

Figure 6 presents a comprehensive evaluation of mixing performance across flow rates. At 20 $mL\cdot {min}^{-1}$ (Figure 6a), AFR-1 initially demonstrates superior mixing during stages s0–s2, attributable to its confined geometry enhancing diffusive mixing. However, beyond stage s3, DDR achieves dominance, reaching complete mixing by s5 through its optimized chaotic convection patterns—a 40% reduction in required mixing cells compared to AFR-1. This transition highlights DDR’s unique capability to convert laminar flow energy into efficient mixing through its concave-induced vortices (Figure 4a,d,g).

The flow rate dependence reveals DDR’s distinct advantage: while all reactors show improved mixing with an increasing flow rate (Figure 6b–c), DDR exhibits exceptional scaling behavior. At 40 $mL\cdot {min}^{-1}$, it achieves 90% mixing by s3 (Figure 6b), outperforming AFR-1 by 13% and AFR-2 by 50% at equivalent stages. This enhancement stems from activated Dean vortices that become prominent above critical flow rates, as evidenced by the streamline analysis in Figure 4d.

The pressure characteristics analysis (Figure 6d) reveals DDR’s exceptional balance between performance metrics. Notably, the design maintains a pressure reduction compared to AFR-1 across all flow rates, especially about a 70% reduction versus AFR-1 at 60 mL·min^−1^ (18 kPa vs. 60 kPa in Figure 6d), while achieving superior mixing efficiency. At flow rates where AFR-2 demonstrates limited mixing performance (<65% mixing index), DDR consistently maintains >90% mixing efficiency. Furthermore, DDR accomplishes complete mixing with less energy input than AFR-1, demonstrating remarkable energy efficiency.

A comparative analysis shows that, while AFR-1’s narrow geometry enhances low-flow mixing, this advantage comes at the cost of significantly higher pressure drops. Conversely, AFR-2’s expanded channels effectively reduce pressure but substantially compromise mixing quality. DDR’s innovative design successfully overcomes this traditional trade-off between mixing performance and pressure characteristics.

The comprehensive dataset demonstrates DDR’s superior operational performance, delivering higher mixing efficiency than AFR-2 at equivalent flow rates while maintaining lower pressure drop than AFR-1 at 60 $mL\cdot {min}^{-1}$. The microreactor exhibits stable operation across the entire tested range of 5–60 $mL\cdot {min}^{-1}$, making it particularly suitable for scalable nanoparticle synthesis applications that demand both high throughput and precise mixing control.

### 3.2. Validation on Mixing Performance by Villermaux-Dushman Experiments

The Villermaux–Dushman experimental results (Figure 7) quantitatively demonstrate DDR’s superior mixing performance. Evaluating the mixing performance of microreactors through energy consumption is a practical method that is widely adopted within the industry. There is a defined energy consumption, as follows [11]:$$\varepsilon =\frac{Q∆P}{\rho V}$$
where *Q* is the total volume flow rate, Δ*P* is the pressure drop, *V* is the volume of the micromixer, and ρ is the density of the mixed solution. ε compares the energy dissipation rate, which increases with the total flow rate. As shown in Figure 7, the Villermaux–Dushman reaction experiments reveal the influence of the energy dissipation rate on the segregation index, further confirming the superior mixing performance of DDR compared to AFR-1, with segregation indices $(X_{s}$) ranging from 0.15 down to 0.027. Specifically, Figure 7c demonstrates that DDR achieves its optimal mixing performance at an energy dissipation rate of 101 W/kg, corresponding to a remarkably low segregation index of 0.027. More importantly, under identical energy input conditions, DDR consistently achieves lower segregation indices than AFR-1 across both low- and high-power regimes. This indicates that DDR delivers higher energy utilization efficiency in pursuit of optimal mixing performance.

Visual observations corroborate these quantitative measurements of product solutions (Figure 7a,b), transitioning from light yellow at low flow rates (indicating higher $I_{2}$ generation) to colorless at elevated flow rates.

This flow-dependent behavior stems from competing reaction kinetics: at lower flow rates, the slower Reaction (6) dominates, yielding more $I_{2}$ and consequently higher $X_{s}$ values. As flow rates increase, enhanced mixing promotes the quasi-instantaneous Reaction (5), effectively consuming available protons before they can participate in $I_{2}$ formation. The DDR’s superior performance is attributed to its optimized geometry that generates intense chaotic advection, ensuring more efficient proton distribution and consumption compared to the AFR-1 design.

### 3.3. Validation on Capability of Continuous Synthesis of BaSO_4_ Nanoparticles

Building upon the comprehensive analysis of flow dynamics and mixing performance in DDR, AFR-1, and AFR-2 microreactors, we next evaluated their effectiveness for continuous nanoparticle synthesis with physical tests. The comparative assessment focused on two key operational parameters, namely total flow rate and reactant concentration, examining their influence on both average particle size and size distribution (PSD). XRD characterization (Figure 8) confirmed the successful synthesis of phase-pure BaSO_4_ nanoparticles in both reactor configurations, with all diffraction peaks precisely matching the reference pattern (JCPDS NO. 24-1035). This crystallographic verification demonstrates that, while both reactor types produce chemically pure products, their differing mixing mechanisms yield distinct morphological outcomes, as revealed by subsequent SEM analysis.

#### 3.3.1. Effect of Flow Rate

The systematic evaluation of BaSO_4_ nanoparticle synthesis reveals distinct performance characteristics between DDR and AFR-1 microreactors, as evidenced by multiple characterization techniques. SEM analysis (Figure 9, Figure 10 and Figure 11) reveals that both microreactors are capable of producing nanoparticles whose average diameter decreases with an increasing flow rate. DDR achieves significantly better size control across all tested flow rates (20–100 $mL\cdot {min}^{-1}$). Figure 10 and Figure 11 present the particle size distributions observed in AFR-1 and DDR under identical initial reactant concentrations, where variations in total flow rate alter the flow velocity, thereby influencing mass transfer, micromixing, and the resulting particle precipitation size. In microreactors, high concentrations of precursors give rise to highly supersaturated product solutions at the interface between the two liquid phases. Following a nucleation burst, the nuclei are carried downstream into regions of comparatively lower solute concentration. In these regions, solute species reach the crystal surfaces via diffusion or convection, and the crystal growth rate is strongly dependent on the local supersaturation distribution and the mixing efficiency of the surrounding fluid. The observed broad particle size distributions in both microreactors can be attributed to two key factors:(1)Crystal nuclei are rarely exposed to uniform, isotropic concentration gradients throughout the reactor. In regions with relatively stagnant flow, mixing is dominated by diffusion alone. Under such anisotropic conditions, the crystal growth process becomes heterogeneous, leading to broader particle size distributions;(2)Collisions among crystals, interactions between crystals and reactor walls, and shear forces within the flow field can result in crystal breakage [33]. Furthermore, particle aggregation leads to an even wider distribution in final particle sizes [34].

The particle size distributions in Figure 10f–j and Figure 11f–j clearly show DDR’s superior uniformity, particularly at higher flow rates, where its geometric standard deviation ($\sigma_{g}$ = 1.27 at 100 $mL\cdot {min}^{-1}$) is 14% narrower than AFR-1’s best performance ($\sigma_{g}$ = 1.47). The flow rate dependence presents particularly revealing contrasts:(1)At the lowest flow rate (20 mL · min^−1^), both reactors produce rod-like particles (Figure 10a and Figure 11a), but DDR already shows a 27% smaller average size (373 nm vs. AFR-1’s 507 nm);(2)As the flow increases to 60 mL · min^−1^, DDR’s particles transition to spherical morphology (Figure 11c) while maintaining a tight size distribution ($\sigma_{g}$ = 1.44), whereas AFR-1 shows mixed morphologies with broader distribution (Figure 10h, $\sigma_{g}$ = 1.58);(3)At peak flow (100 mL · min^−1^), DDR achieves its optimal performance with 95 nm spherical particles (Figure 11e,j), while AFR-1 produces 177 nm particles (Figure 10j) with visible aggregation. The literature has shown that in laminar-flow microreactors, higher supersaturation ratios and increased fluid velocities result in smaller particle sizes, while the shear forces within the channel can significantly influence convective mass transfer [35]. A comparison of the streamline patterns between AFR-1 and DDR (Figure 4d,e) reveals that, under identical flow conditions, the hierarchical flow division structure of DDR induces continuous fluid splitting and recombination. This process substantially increases the interfacial shear area between the two liquid phases compared to AFR-1. As a result, BaSO_4_ nanoparticles formed in DDR exhibit a smaller average size (95 nm) than those in AFR-1 (177 nm). Supporting this observation, Sen et al. [36] reported that enhanced mixing leads to a more homogeneous precursor concentration field, which helps maintain a consistent reaction rate and yields smaller, more uniform particles. The smaller average particle size produced in DDR further underscores its superior mixing performance compared to AFR-1.

These morphological differences correlate directly with the mixing characteristics revealed in Figure 4, Figure 5 and Figure 6:(1)The velocity contours (Figure 4a) show DDR’s more uniform flow field, which eliminates the stagnant zones observed in AFR-1 (Figure 4b);(2)Mixing index profiles (Figure 6a–c) demonstrate DDR’s faster approach to complete mixing (reaching 90% by stage s3 at 40 $mL\cdot {min}^{-1}$);(3)Pressure characteristics (Figure 6d) confirm DDR maintains this superior mixing at a three-times pressure drop than AFR-1 at 60 $mL\cdot {min}^{-1}$.

The XRD patterns (Figure 8) provide important context—both reactors produce phase-pure BaSO_4_, confirming that the performance differences stem from fluid dynamics rather than chemical purity. This is further supported by the Villermaux–Dushman results (Figure 7), where DDR’s lower segregation indices (0.027–0.15 vs. AFR-1’s 0.1–0.4) quantitatively verify its mixing superiority. The combined dataset explains DDR’s advantages through three interlinked mechanisms:(1)Enhanced supersaturation control: DDR’s splitting–recombination design (Figure 1a) creates more uniform concentration fields (Figure 5a), promoting homogeneous nucleation via the LaMer mechanism [37];(2)Reduced growth dominance: The concave structures (Figure 4a) generate vortices that minimize particle residence in growth-favoring zones, suppressing Ostwald ripening;(3)Efficient energy utilization: As shown in Figure 6d, DDR achieves better mixing at a lower energy input, preventing the overgrowth seen in AFR-1 products.

The comprehensive characterization demonstrates that DDR’s design innovations translate to measurable improvements in nanoparticle synthesis:(1)46% smaller average size at 100 $mL\cdot {min}^{-1}$ flow rate (95 nm vs. AFR-1’s 177 nm, as shown in Figure 9, Figure 10e and Figure 11e);(2)14% narrower size distributions at 100 $mL\cdot {min}^{-1}$ flow rate (*σ_g_* = 1.27 vs. 1.47, as shown in Figure 11j vs. Figure 10j);(3)Faster morphological transition to desirable spherical shapes (60 mL·min^−1^ vs. 80 mL·min^−1^, as shown in Figure 11c vs. Figure 10c);(4)Better scaling with increasing flow rate, evidenced by an about 2.8 nm·(mL/min)^−1^ size reduction rate in Figure 9.

These advantages, corroborated by both fluid dynamics analysis and product characterization, position DDR as superior for precision nanoparticle synthesis where control over both size and morphology is critical. The consistent correlation between mixing performance (Figure 4, Figure 5 and Figure 6) and product characteristics (Figure 9, Figure 10 and Figure 11) validates the importance of microreactor design in controlling nucleation and growth processes.

#### 3.3.2. Effect of Reactant Concentration

In this section, the effect of precursor concentration on the preparation of BaSO_4_ nanoparticles in AFR-1 and DDR was examined. The conditions were as follows: the total flow rate was 100 $mL\cdot m^{-1}$, the flux ratio of the two reactants was 1, and $C\left({Ba}^{2+}\right)$ varied from 0.1 to 1 mol/L.

The results for the SEM images are illustrated in Figure 12 and Figure 13, and Figure 14 shows the trend of the particle size with various reactant concentrations. As depicted in Figure 13 and Figure 14, the samples prepared via AFR-1 and DDR exhibit a similar phenomenon in that the products show a 2D plate-like morphology at a low initial reactant concentration (0.1 mol/L), whereas 3D spherical crystals are synthesized when the initial reactant concentration increases to 0.3 mol/L. Figure 12 indicates an obvious decrease in the average particle diameter when the precursor concentration increases from 0.1 mol/L to 0.3 mol/L, while the diameter decreases as the precursor concentration increases from 0.3 mol/L to 0.5 mol/L. Furthermore, regardless of the initial precursor concentration, BaSO_4_ nanoparticles prepared through DDR are smaller than those synthesized by AFR-1.

A higher initial precursor concentration leads to a higher reaction rate, which increases both the nucleation and growth rates. The size of the nanoparticle in a precipitation process is controlled by the supersaturation ratio. With a high supersaturation ratio, nucleation bursts at the fluid interface dominate small particle generation, while a low supersaturation ratio generates larger samples for particle growth [38]. Judat B. et al. [39] demonstrated that particle morphology is predominantly determined by the initial supersaturation ratio, with increased supersaturation promoting the formation of more uniformly shaped crystals. Abdel-Aal et al. [40] observed that high supersaturation levels yielded larger crystals, while lower supersaturation resulted in the formation of thicker but less extensive crystals. Similarly, Hu et al. [41] reported that decreasing the supersaturation led to a reduction in the crystal size of aragonite. Therefore, a low initial precursor concentration leads to a low supersaturation ratio at the liquid-liquid interface, where the nuclei are synthesized. The further process of growth shapes the nuclei into tabular crystals. This phenomenon indicates that a low initial precursor concentration is beneficial for growth rather than nucleation. The action of the shear force between two phases in front of the splitter of the DDR generates a strong inner circulatory flow, and the mass transfer speed is very high, which leads to an increase in the supersaturation ratio [42]. Thus, the average particle diameter of BaSO_4_ synthesized via DDR is smaller than that of BaSO4 synthesized through AFR-1, implying that DDR has a better mixing performance.

Our investigation of precursor concentration effects (0.1–1 mol/L ${Ba}^{2+}$) at fixed flow conditions (100 $mL\cdot m^{-1}$) reveals fundamental differences in crystallization behavior between reactor designs, as clearly demonstrated in Figure 12, Figure 13 and Figure 14. The SEM analyses show three distinct concentration regimes:(1)Low concentration regime (0.1 mol/L): The SEM images in Figure 13a and Figure 14a reveal that both reactor configurations produce 2D plate-like crystals at the lowest concentration, though with distinct size distributions. DDR generates notably smaller nanoplates, with an average size of 719 nm (Figure 14f), compared to the 1929 nm plates from AFR-1 (Figure 13f). This size difference becomes visually apparent when comparing the edge clarity in Figure 14a (DDR) vs. Figure 13a (AFR-1), where DDR’s plates show more uniform electron contrast. The mixing patterns shown in Figure 5a explain this advantage—DDR’s concave structures create more uniform concentration gradients during the initial crystallization phase, preventing the localized overgrowth observed in AFR-1’s more turbulent flow field (Figure 5b);(2)Transition concentration (0.3 mol/L): At this critical concentration, both reactors undergo a morphological transition, but with different kinetics and outcomes. DDR completes the shift to 3D spherical particles (Figure 14b) with an average diameter of 123 nm, while AFR-1 produces a mixed population of spheres (330 nm) and residual plates (Figure 14a). The velocity profiles in Figure 4 demonstrate why DDR achieves cleaner transitions—its stable vortical flows (Figure 4a) maintain consistent supersaturation, whereas AFR-1’s U-shaped obstacles (Figure 4b) create fluctuating conditions that allow both growth mechanisms to coexist;(3)High concentration regime (0.5–1 mol/L): In this regime, the reactors exhibit fundamentally different scaling behaviors. DDR maintains a spherical morphology across all high concentrations (Figure 14c–e), with particle sizes stabilizing at 95 nm (0.5 mol/L) and showing minimal further reduction at higher concentrations. In contrast, AFR-1 displays continued size variation (177 nm at 0.5 mol/L). The mixing index data in Figure 6 explains this divergence—DDR maintains a >90% mixing efficiency throughout the concentration range, while AFR-1’s performance degrades due to increased viscosity effects in its narrower channels. Remarkably, DDR achieves this while maintaining a lower pressure drop than AFR-1 (Figure 6d), demonstrating its energy efficiency advantage for concentrated precursor solutions.

## 4. Conclusions

This study introduces a novel planar microreactor design (DDR) and systematically evaluates its performance against two current state-of-the-art configurations (AFR-1 and AFR-2) through combined computational and experimental approaches. The key findings demonstrate the superiority of the innovative DDR design.

Computational fluid dynamics (CFD) simulations reveal that DDR’s unique splitting–turning–impinging architecture achieves exceptional mixing via synergistic mechanisms: (i) periodic fluid segmentation maximizing the interfacial area, (ii) generation of controlled vortical flows inducing efficient chaotic advection, and (iii) maintenance of stable pressure characteristics (e.g., <20 kPa at 60 mL·min^−1^), consistently delivering mixing indices >90% at the outlet. These mechanisms collectively outperform those observed in the benchmark advanced configurations.

Quantitative Villermaux–Dushman experiments provide conclusive evidence of DDR’s enhanced mixing relative to the advanced benchmarks, yielding significantly lower segregation indices ($X_{s}$ = 0.027 at 100 mL·min^−1^) compared to AFR-1 ($X_{s}$ = 0.0985). Crucially, this mixing advantage translates directly into superior nanoparticle synthesis outcomes. DDR produced BaSO_4_ nanoparticles with a 46% smaller average diameter (95 nm vs. 177 nm for AFR-1 at 100 mL·min^−1^), a 14% narrower size distribution (*σ_g_* = 1.27 vs. 1.47), and a more uniform spherical morphology, even at significantly lower precursor concentrations (0.3 mol/L vs. 0.5 mol/L required by AFR-1).

Beyond its core mixing and synthesis performance, DDR offers compelling practical advantages over the advanced benchmarks: (i) exceptional operational stability over a broader flow range (5–100 mL·min^−1^), (ii) lower pressure drops across comparable flow rates, implying reduced energy consumption, and (iii) a scalable architecture proven to maintain high performance consistency during continuous nanoparticle production.

The strong correlation between DDR’s novel design-driven mixing efficiency and the demonstrably improved nanoparticle quality validates its innovative principles. These advantages position DDR as a promising platform, particularly for applications demanding high-throughput nanoparticle synthesis, precise size and morphology control, and energy-efficient operation, outperforming current advanced designs. The reactor’s performance superiority is consistently evidenced across all characterization techniques, from CFD to experimental analysis.

Future research should focus on (i) geometric optimization of the DDR design to further enhance performance, (ii) comprehensive scale-up studies for industrial deployment, and (iii) extending the DDR methodology to diverse precipitation systems. The success of DDR’s design philosophy, combining controlled flow segmentation with chaotic advection, establishes a valuable framework for developing advanced microreactors tailored for precision nanomaterial manufacturing.

## Figures and Tables

**Figure 1 micromachines-16-01058-f001:**
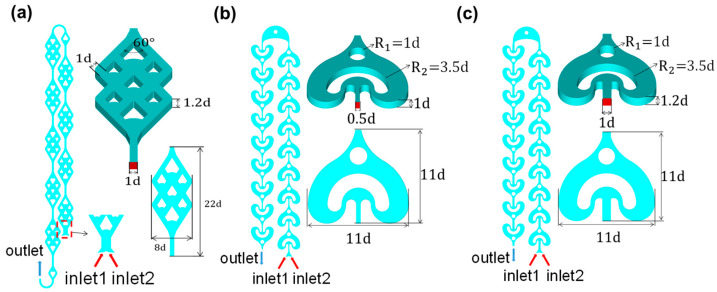
(**a**) The double-diamond-shaped reactor (DDR). (**b**) The advanced flow reactor 1 (AFR-1). (**c**) The advanced flow reactor 2 (AFR-2).

**Figure 2 micromachines-16-01058-f002:**
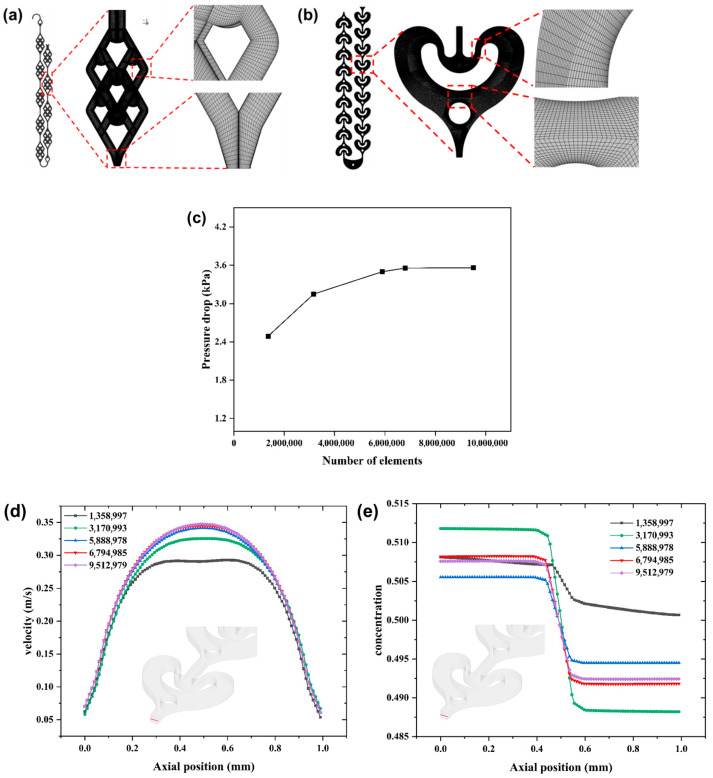
(**a**) Mesh of DDR. (**b**) Mesh of AFR-1; grid independence check: (**c**) pressure drop, (**d**) velocity distribution curve at the outlet tangent line, (**e**) normalized concentration (range 0–1) distribution curve at the outlet tangent line.

**Figure 3 micromachines-16-01058-f003:**
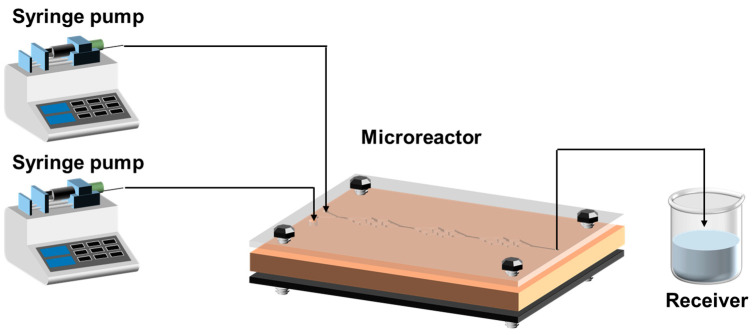
Schematic of the experimental setup for the synthesis of BaSO_4_ nanoparticle.

**Figure 4 micromachines-16-01058-f004:**
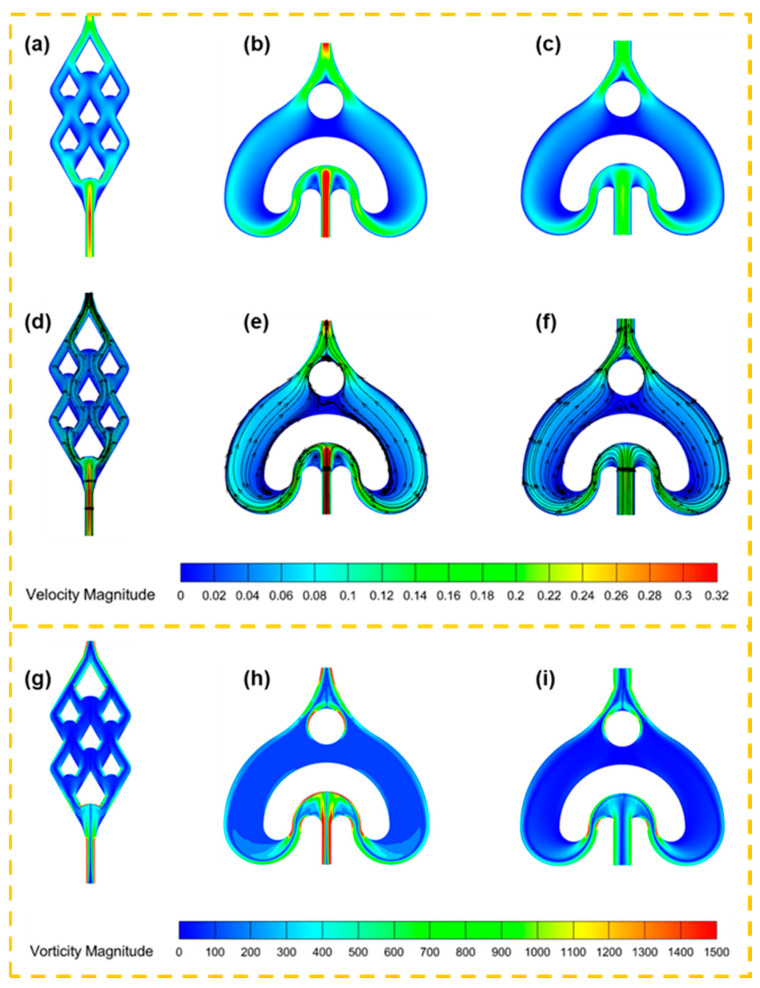
(**a**–**c**) Velocity contours, (**d**–**f**) streamlines, and (**g**–**i**) vorticity contours in the X–Z plane at the middle of the channel depth when the total flow rate is 10 $mL\cdot {min}^{-1}$, (**a**,**d**,**g**) DDR, (**b**,**e**,**h**) AFR-1, (**c**,**f**,**i**) AFR-2.

**Figure 5 micromachines-16-01058-f005:**
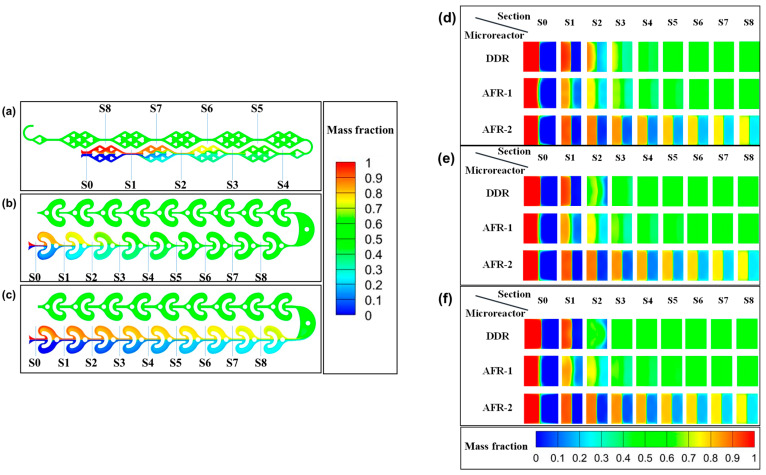
Different longitudinal sections of mass fraction contours in (**a**) DDR, (**b**) AFR-1, and (**c**) AFR-2 at different total flow rates: (**d**) 20 $mL\cdot {min}^{-1}$, (**e**) 40 $mL\cdot {min}^{-1}$, (**f**) 60 $mL\cdot {min}^{-1}$.

**Figure 6 micromachines-16-01058-f006:**
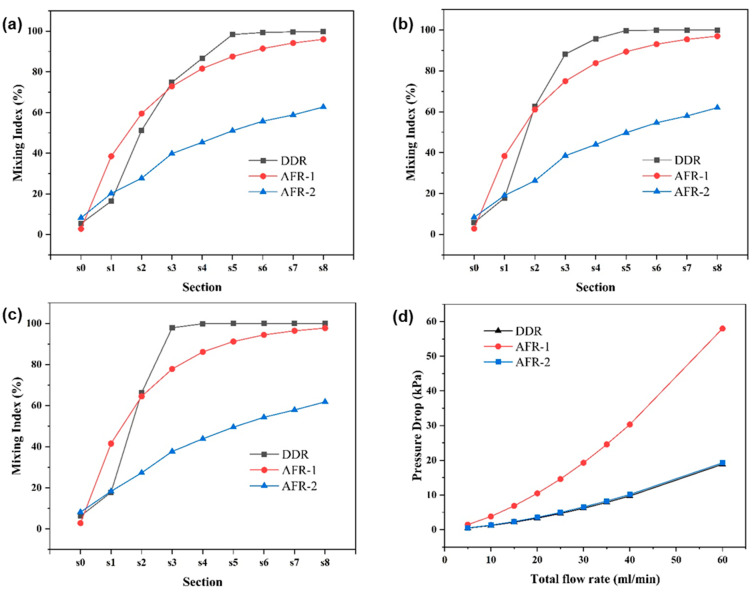
Mixing index (*MI*) of longitudinal sections in three microreators at different total flow rates: (**a**) 20 $mL\cdot {min}^{-1}$, (**b**) 40 $mL\cdot {min}^{-1}$, (**c**) 60 $mL\cdot {min}^{-1}$, and (**d**) the pressure drops with different total flow rates.

**Figure 7 micromachines-16-01058-f007:**
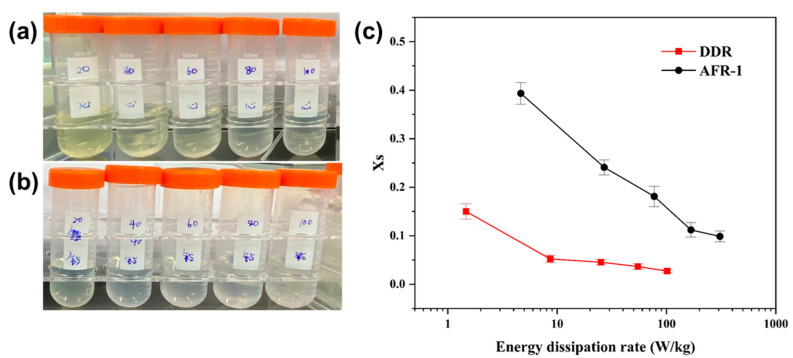
V-D experiment for AFR-1 versus DDR at the total flow rate range from 20 to 100 $mL\cdot {min}^{-1}$. (**a**) Product samples of Villermuax–Dushman experiment for AFR-1. (**b**) Product samples of Villermuax–Dushman experiment for DDR. (**c**) The effects of energy dissipation rate on segregation index in DDR and AFR-1.

**Figure 8 micromachines-16-01058-f008:**
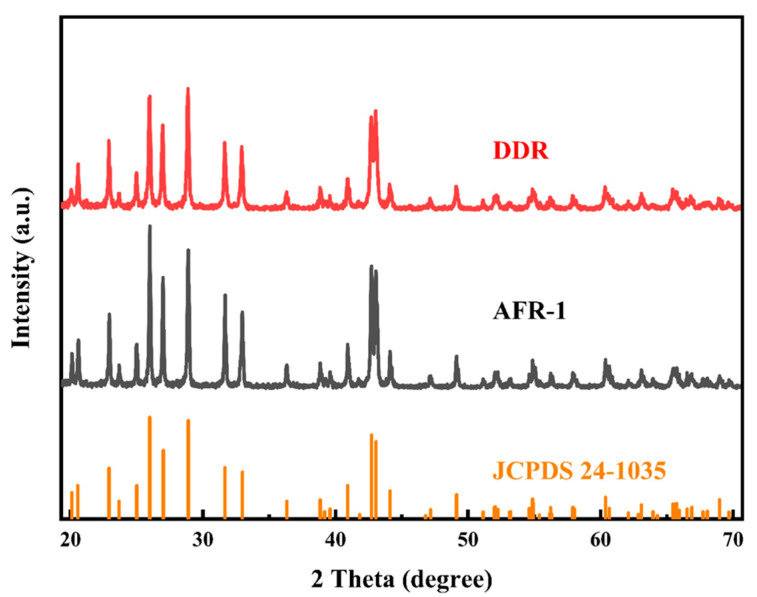
XRD pattern of BaSO_4_ samples prepared by the two microreactors.

**Figure 9 micromachines-16-01058-f009:**
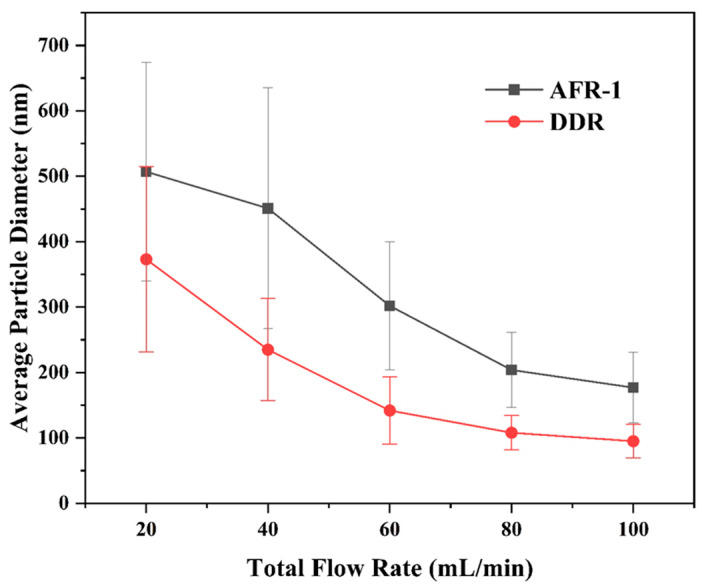
The average particle diameter of BaSO_4_ varies with varying total flow rate in two microreactors based on SEM images $:C\left({Ba}^{2+}\right)=C\left({SO}_{4}^{2-}\right)=0.5$ mol/L.

**Figure 10 micromachines-16-01058-f010:**
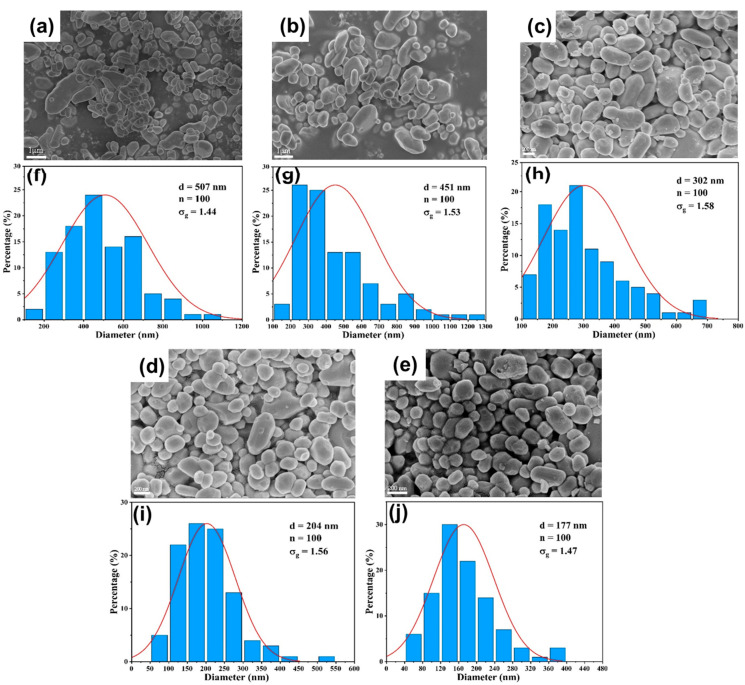
(**a**–**e**) SEM images and (**f**–**j**) particle size distribution of BaSO_4_ nanoparticles at different flow rates in AFR-1: (**a**,**f**) $Q_{total}=20\;mL\cdot {min}^{-1}$; (**b**,**g**) $Q_{total}=40\;mL\cdot {min}^{-1}$; (**c**,**h**) $Q_{total}=60\;mL\cdot {min}^{-1}$; (**d**,**i**) $Q_{total}=80\;mL\cdot {min}^{-1}$; (**e**,**j**) $Q_{total}=100\;mL\cdot {min}^{-1}$; $C\left({Ba}^{2+}\right)=C\left({SO}_{4}^{2-}\right)=0.5\;mol/L$.

**Figure 11 micromachines-16-01058-f011:**
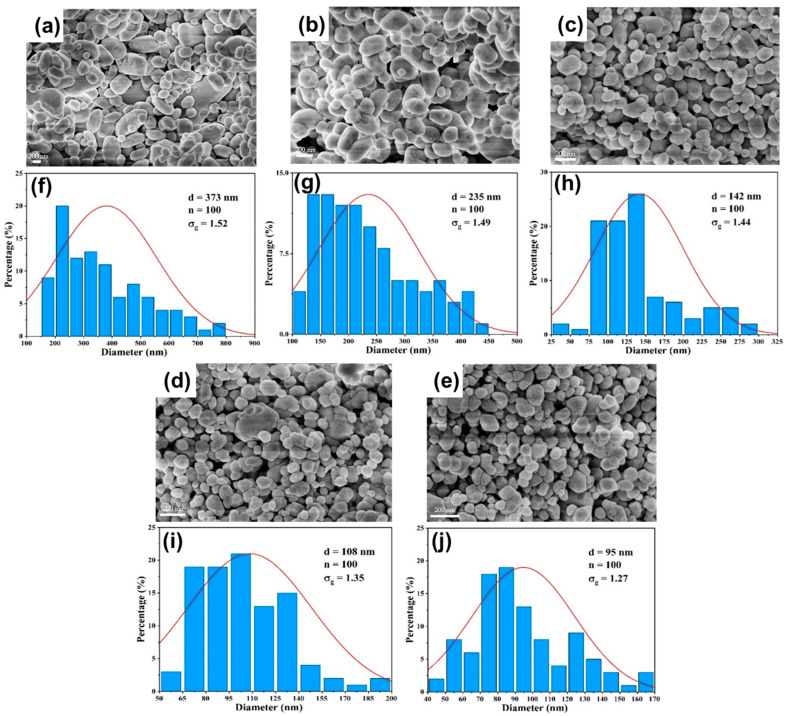
(**a**–**e**) SEM images and (**f**–**j**) particle size distribution of BaSO_4_ nanoparticles at different flow rates in DDR: (**a**,**f**) $Q_{total}=20\;mL\cdot {min}^{-1}$; (**b**,**g**) $Q_{total}=40\;mL\cdot {min}^{-1}$; (**c**,**h**) $Q_{total}=60\;mL\cdot {min}^{-1}$; (**d**,**i**) $Q_{total}=80\;mL\cdot {min}^{-1}$; (**e**,**j**) $Q_{total}=100\;mL\cdot {min}^{-1}$; $C\left({Ba}^{2+}\right)=C\left({SO}_{4}^{2-}\right)=0.5\;mol/L$.

**Figure 12 micromachines-16-01058-f012:**
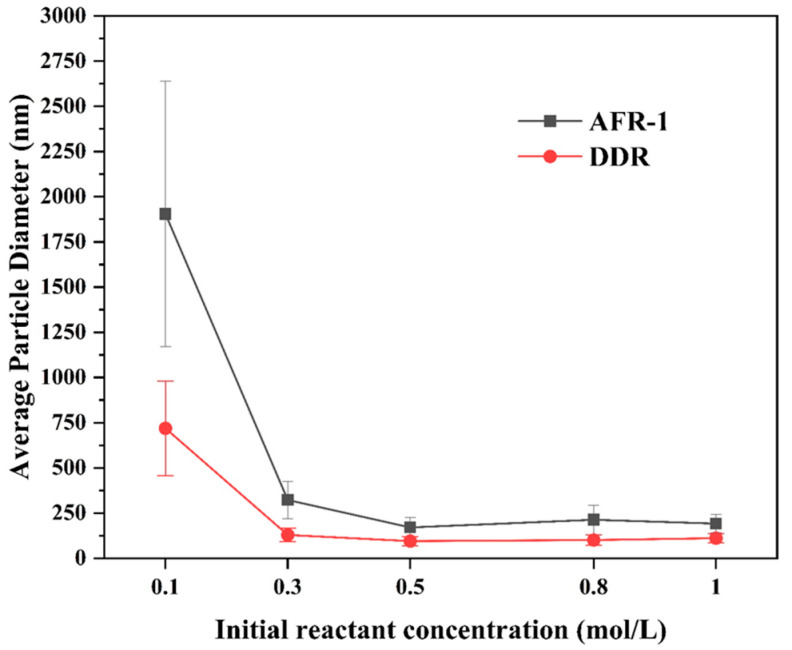
The average particle diameter of BaSO_4_ varies with varying precursor concentration in two microreactors, based on SEM images $:Q_{total}=100\;mL\cdot {min}^{-1}$.

**Figure 13 micromachines-16-01058-f013:**
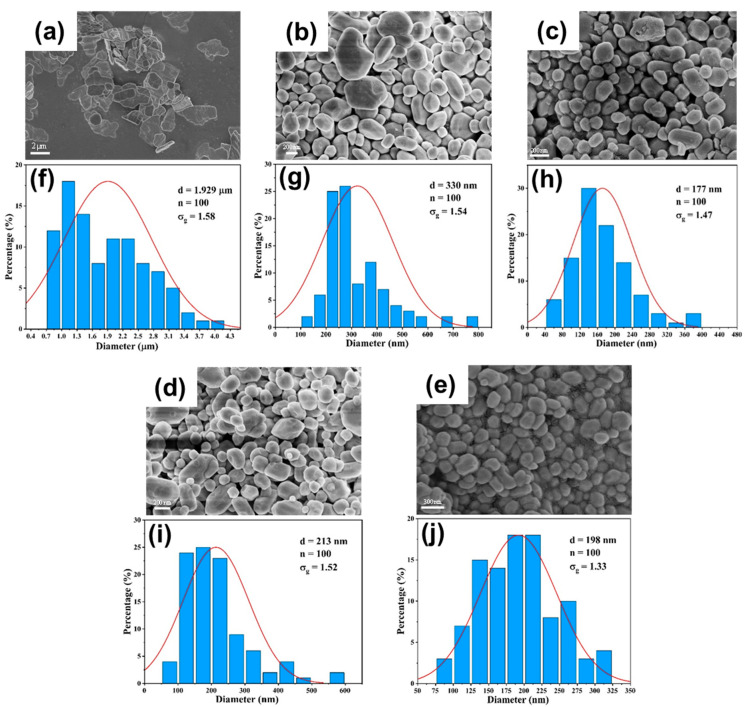
(**a**–**e**) SEM images and (**f**–**j**) particle size distribution of BaSO_4_ nanoparticles at different precursor concentrations in AFR-1: (**a**,**f**) $C\left({Ba}^{2+}\right)=C\left({SO}_{4}^{2-}\right)=0.1\;mol/L$; (**b**,**g**) $C\left({Ba}^{2+}\right)=C\left({SO}_{4}^{2-}\right)=0.3\;mol/L$; (**c**,**h**) $C\left({Ba}^{2+}\right)=C\left({SO}_{4}^{2-}\right)=0.5\;mol/L$; (**d**,**i**) $C\left({Ba}^{2+}\right)=C\left({SO}_{4}^{2-}\right)=0.8\;mol/L$; (**e**,**j**) $C\left({Ba}^{2+}\right)=C\left({SO}_{4}^{2-}\right)=1\;mol/L$; $Q_{total}=100\;mL\cdot {min}^{-1}$.

**Figure 14 micromachines-16-01058-f014:**
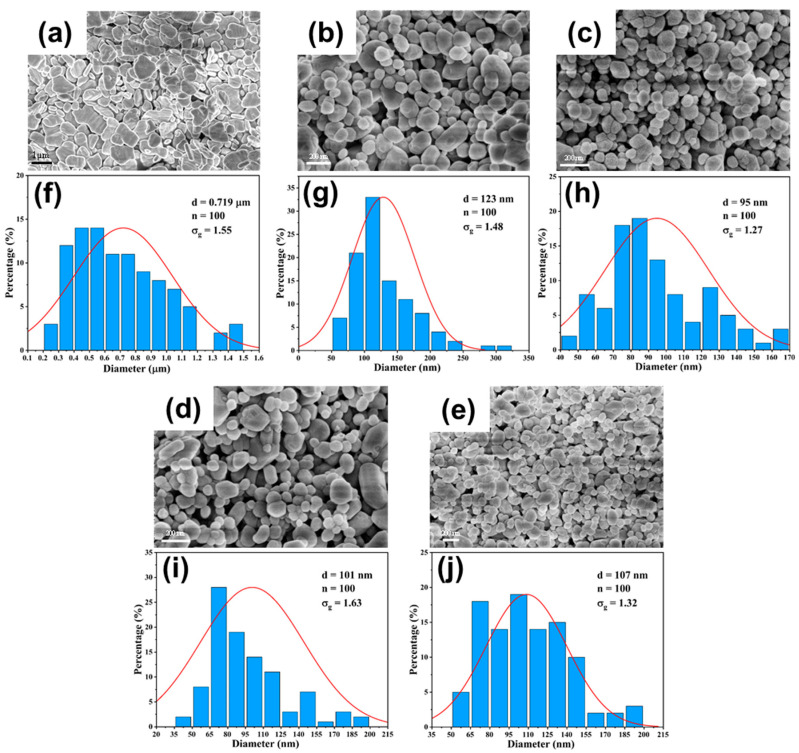
(**a**–**e**) SEM images and (**f**–**j**) particle size distribution of BaSO_4_ nanoparticles at different precursor concentrations in DDR: (**a**,**f**) $C\left({Ba}^{2+}\right)=C\left({SO}_{4}^{2-}\right)=0.1\;mol/L$; (**b**,**g**) $C\left({Ba}^{2+}\right)=C\left({SO}_{4}^{2-}\right)=0.3\;mol/L$; (**c**,**h**) $C\left({Ba}^{2+}\right)=C\left({SO}_{4}^{2-}\right)=0.5\;mol/L$; (**d**,**i**) $C\left({Ba}^{2+}\right)=C\left({SO}_{4}^{2-}\right)=0.8\;mol/L$; (**e**,**j**) $C\left({Ba}^{2+}\right)=C\left({SO}_{4}^{2-}\right)=1\;mol/L$; $Q_{total}=100\;mL\cdot {min}^{-1}$.

**Table 1 micromachines-16-01058-t001:** Kinetic parameters of the Villermaux–Dushman reaction [27].

Reaction	Kinetics	k
Reaction (5) [29]	$r_{1}=k_{1}\left[H^{+}\right][H_{2}BO_{3}^{-}]$	$k_{1}=10^{11}\;m^{3}$/(kmol·s)
Reaction (6) [30]	$r_{2}=k_{2}\left[IO_{3}^{-}\right]{\left[I^{-}\right]}^{2}{\left[H^{+}\right]}^{2}$	$k_{2}=4.27\times 10^{8}\;m^{12}$/$\left({kmol}^{4}\cdot s\right)$
Reaction (7) [30]	$r_{3}=k_{3}\left[I^{-}\right]\left[I_{2}\right]-k_{4}[I_{3}^{-}]$	$k_{3}=5.90\times 10^{9}\;m^{13}$/(kmol·s)
		$k_{4}=7.50\times 10^{9}\;s^{-1}$

## Data Availability

Data will be made available on request.

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
