# Peer review of "A Novel Double-Diamond Microreactor Design for Enhanced Mixing and Nanomaterial Synthesis"

_micromachines, 2025, doi:10.3390/mi16091058_

Round 1
Reviewer 1 Report
Comments and Suggestions for Authors
- Lines 157-161: Please discuss mesh non-uniformity. How does mesh non-uniformity affect flow and mixing?
- Fig.2 (c): Compare results with the measured data for DDR and discuss
- Fig. 2 (c): Both the flow and mixing should pass the mesh independence check. Correct.
- Lines 282-298: Explain the novelty in the definition of the mixing index, compare it with the classical approach, and highlight the advantages of the new expression over the traditional definition. Please include reference to the conventional definition of MI.
- Fig. 7(c): Alternatively, the direct effect of the total power consumption per unit mass on XS should be determined. Both reactors (AFR-1 and DDR) should be compared, and the results comprehensively discussed.
- Lines 403-405 (visible aggregation): Explain in more detail the role of mixing and supersaturation in the formation or prevention of crystal aggregates. Refer to previously published data on BaSO4 aggregation.
- Lines 470-474: Such observations were made well before 2004; cite more fundamental studies on the influence of micromixing and supersaturation ratio on the crystal size and its morphology.
Reviewer 2 Report
Comments and Suggestions for Authors
This manuscript presented a Double-Diamond Reactor (DDR) design and utilized the design for mixing and nanomaterial synthesis. The simulation and experiment results confirmed that this design showed better performance than reference design (AFR). Overall, the idea was interesting. There are a few suggests or questions as follows:
- It is recommended to include additional recent references, as some of the current citations are outdated.
- Error bars should be incorporated into Fig. 9 and Fig.12 to more clearly represent the size distribution of the nanoparticles.
- P13, Line 390, the statement “both reactors produce nanoparticles in the 95-507 nm range” is not accurate. While the average nanoparticle sizes were controlled between 95 and 507 nm, the actual size distribution of the generated nanoparticles extends beyond these bounds.
- Regarding Fig.10 and Fig.11, please provide a discussion on the possible factors contributing to the broad size distribution observed for BaSO4 nanoparticles.
Minor comments:
(1) Please verify and correct any font formatting errors on P3, Line 93-96.
(2) Please check formatting errors in this manuscript. For example, P7, Line209, “BaSO4 nanoparticles is shown in Fig.4”. And the schematic was shown in Fig.3.
(3) It is suggested to reorganize the layout of Fig.5. Fig.5 (a)-(c) could be arranged horizontally for improved clarity.
(4) The annotation for Fig.7 (c) is missing in Fig.7. And the labeling styles of panels (a), (b), and (c) is inconsistent and visually unappealing. These could be standardized and improved.
Round 2
Reviewer 1 Report
Comments and Suggestions for Authors
no further comments or suggestions